# SARS-CoV-2 Proteins Bind to Hemoglobin and Its Metabolites

**DOI:** 10.3390/ijms22169035

**Published:** 2021-08-21

**Authors:** Guilherme C. Lechuga, Franklin Souza-Silva, Carolina Q. Sacramento, Monique R. O. Trugilho, Richard H. Valente, Paloma Napoleão-Pêgo, Suelen S. G. Dias, Natalia Fintelman-Rodrigues, Jairo R. Temerozo, Nicolas Carels, Carlos R. Alves, Mirian C. S. Pereira, David W. Provance, Thiago M. L. Souza, Salvatore G. De-Simone

**Affiliations:** 1FIOCRUZ, Center for Technological Development in Health (CDTS), National Institute of Science and Technology for Innovation on Neglected Population Diseases (INCT-IDPN), Rio de Janeiro 21040-900, RJ, Brazil; guilherme.lechuga@cdts.fiocruz.br (G.C.L.); franklin.frankss@gmail.com (F.S.-S.); carol.qsacramento@gmail.com (C.Q.S.); mrotrugilho@hotmail.com (M.R.O.T.); napoleaopego@cdts.fiocruz.br (P.N.-P.); nataliafintelman@gmail.com (N.F.-R.); nicolas.carels@cdts.fiocruz.br (N.C.); bill.provance@cdts.fiocruz.br (D.W.P.J.); thiago.moreno@cdts.fiocruz.br (T.M.L.S.); 2Laboratory of Celular Ultrastructure, FIOCRUZ, Oswaldo Cruz Institute, Rio de Janeiro 21040-900, RJ, Brazil; mirian@ioc.fiocruz.br; 3Biology and Heath Science Faculty, Iguaçu University, Nova Iguaçu 26260-045, RJ, Brazil; 4Laboratory of Immunopharmacology, FIOCRUZ, Oswaldo Cruz Institute, Rio de Janeiro 21040-900, RJ, Brazil; suelen_sgdias@hotmail.com; 5Laboratory of Toxinology, FIOCRUZ, Oswaldo Cruz Institute, Rio de Janeiro 21040-900, RJ, Brazil; richardhemmi@gmail.com; 6Laboratory of Thymus Research, FIOCRUZ, Oswaldo Cruz Institute, Rio de Janeiro 21040-900, RJ, Brazil; jairo.jrt@gmail.com; 7FIOCRUZ, National Institute for Science and Technology on Neuroimmunomodulation (INCT/NIM), Rio de Janeiro 21040-900, RJ, Brazil; 8Laboratory of Molecular Biology and Endemic Diseases, FIOCRUZ, Oswaldo Cruz Institute, Rio de Janeiro 21040-900, RJ, Brazil; calves@ioc.fiocruz.br; 9Department of Cellular and Molecular Biology, Biology Institute, Federal Fluminense University, Niterói 24020-141, RJ, Brazil

**Keywords:** SARS-CoV-2, COVID-19, protein–protein binding, hemoglobin, hemin, RBD, N, S, M, Nsp3, Nsp7

## Abstract

(1) Background: coronavirus disease 2019 (COVID-19), caused by severe acute respiratory syndrome coronavirus 2 (SARS-CoV-2), has been linked to hematological dysfunctions, but there are little experimental data that explain this. Spike (S) and Nucleoprotein (N) proteins have been putatively associated with these dysfunctions. In this work, we analyzed the recruitment of hemoglobin (Hb) and other metabolites (hemin and protoporphyrin IX-PpIX) by SARS-Cov2 proteins using different approaches. (2) Methods: shotgun proteomics (LC–MS/MS) after affinity column adsorption identified hemin-binding SARS-CoV-2 proteins. The parallel synthesis of the peptides technique was used to study the interaction of the receptor bind domain (RBD) and N-terminal domain (NTD) of the S protein with Hb and in silico analysis to identify the binding motifs of the N protein. The plaque assay was used to investigate the inhibitory effect of Hb and the metabolites hemin and PpIX on virus adsorption and replication in Vero cells. (3) Results: the proteomic analysis by LC–MS/MS identified the S, N, M, Nsp3, and Nsp7 as putative hemin-binding proteins. Six short sequences in the RBD and 11 in the NTD of the spike were identified by microarray of peptides to interact with Hb and tree motifs in the N protein by in silico analysis to bind with heme. An inhibitory effect in vitro of Hb, hemin, and PpIX at different levels was observed. Strikingly, free Hb at 1mM suppressed viral replication (99%), and its interaction with SARS-CoV-2 was localized into the RBD region of the spike protein. (4) Conclusions: in this study, we identified that (at least) five proteins (S, N, M, Nsp3, and Nsp7) of SARS-CoV-2 recruit Hb/metabolites. The motifs of the RDB of SARS-CoV-2 spike, which binds Hb, and the sites of the heme bind-N protein were disclosed. In addition, these compounds and PpIX block the virus’s adsorption and replication. Furthermore, we also identified heme-binding motifs and interaction with hemin in N protein and other structural (S and M) and non-structural (Nsp3 and Nsp7) proteins.

## 1. Introduction

Coronavirus disease 2019 (COVID-19), caused by severe acute respiratory syndrome coronavirus 2 (SARS-CoV-2), was first detected in Wuhan (Hubei province, China) at the end of 2019. The disease rapidly spread across the world due to its high transmissibility, prolonged incubation, and a highly connected global travel network [1]. The ongoing COVID-19 pandemic has proven to be a significant economic and public health challenge. The elderly and individuals with pre-existing comorbidities are severely affected, but severe COVID-19 complications can impact any age group [2]. The global scientific community has exerted tremendous effort to understand the viral structure and physiopathology of COVID-19, to identify control measures, including drug repurposing strategies, plasmapheresis, and vaccination [3]. Despite the performance (and increasing availability) of newly developed vaccines—emerging variants of concern, which appear to evade immune responses triggered by vaccines, represent a significant concern to immunization strategies [4].

Drug repurposing has not been unequivocally satisfactory against SARS-CoV-2; however, this strategy could still prove worthy, provided that relevant biochemical interactions of viral proteins and small molecules are determined. Furthermore, by expanding the breadth of knowledge on the activities of individual SARS-CoV-2 proteins during infection, biochemically-based evidence of effective treatments for current and future variants could be accelerated, preventing further frustration in clinical trials with molecules, with limited preclinical effectiveness against SARS-CoV-2 [5].

Hematological COVID-19 is a constitutive component in critically ill patients [6,7]. The heme-iron dysregulation has been observed in COVID-19 [7], with binding signatures including hyperferritinemia, low hemoglobin (Hb) levels, low serum iron, anisocytosis, increased variation of red blood cell distribution width (RDW), and hypoxemia [8,9,10]. Unbalanced erythrocyte, Hb, and iron levels were associated with poor clinical outcomes in COVID-19 [9]. An in silico analysis pointed to a relevant role of SARS-CoV-2 proteins in viral physiopathology. The predictions suggest that the capture of heme, resulting from a coordinated attack of orf1ab, ORF10, and ORF3a to the 1-β chain of hemoglobin, could interfere with heme metabolism and oxygen transport. This analysis also proposed the binding of heme by structural and non-structural proteins of SARS-CoV-2 [6]. Although these data bring interesting perspectives, experimental confirmation is still needed.

Heme, iron protoporphyrin IX (Fe-PpIX), is a ubiquitous molecule with importance in numerous biological processes, such as a cofactor for proteins, transcriptional regulation [11], RNA processing [12], oxidative stress [13], inflammation [14], and coagulation [15], which are all critical aspects of COVID-19 pathology. Heme and porphyrins can modulate viral infection by targeting both viral structures and cellular pathways. Porphyrins have broad activities against different viruses, such as hepatitis B virus (HBV), hepatitis C virus (HCV), human immunodeficiency virus (HIV), and Zika virus (ZIKV) [16,17]. Nonspecific heme interactions, including hydrophobic binding to viral surface envelope proteins, block viral cellular entry [18]. It was recently reported that potent antiviral activity of the heme precursor, PpIX, and verteporfin in the nanomolar range inhibit viral invasion, by blocking the virus-cell fusion mediated by SARS-CoV-2 spike (S) protein and ACE2 [19]. When Vero cells were pretreated with both, there was inhibition of viral RNA production, suggesting that their interactions with ACE2 caused the viral entry block.

Despite concerted efforts to unveil key viral targets, experimental evidence is still limited. Recently, research describing the binding of the products of Hb degradation (bilirubin and biliverdin) to the N-terminus of S protein of SARS-CoV-2 and SARS-CoV1 was published [20]. Here, we show that the S protein and other SARS-CoV-2 structural and non-structural proteins can bind to Hb and the metabolites hemin and PpIX. In addition, in vitro assays showed that free hemin and PpIX precluded at different levels the host cells’ viral attachments to reduce viral replication. This property was mainly associated with the RBD region of the spike protein.

## 2. Results

### 2.1. Identification of the SARS-CoV-2-Hemin Binding Proteins

Two distinct electrophoretic approaches were used to evaluate the cellular and viral proteins’ interaction with heme. In the first approach, the viral proteins (extracts from Vero cell cultures either infected with SARS-CoV-2 or mock-infected, as a control) were previously preincubated with 300 µM hemin and then the proteins resolved by SDS-PAGE (in-gel protein renaturation) and the heme-binding proteins visualized with TMB (3,3′,5,5′-Tetramethylbenzidine) staining. This assay revealed two SARS-CoV-2 bands (ca. 55 and 70 kDa) in the virus-infected Vero cells’ protein extract and a faint band of ~55 kDa in the extract of the mock-infected cell (Figure 1a; right side). As expected, no protein band was visualized in the viral sample without hemin treatment (TMB oxidation) used as a negative control (Figure 1a; left side).

In the second approach, the virus proteins were resolved by SDS-PAGE (under reducing and denaturing conditions), and after blotting to nitrocellulose membrane (NM) was incubated with 2 µM hemin followed by DAB (diaminobenzidine) staining. In this assay, one band with ~55 kDa was prominent, but three other bands (ca. 45, 70, and 230 kDa) were detected exclusively in the extract of SARS-CoV-2-infected cells (Figure 1b). Thus, this approach displayed greater sensitivity for detecting heme-protein complexes. Therefore, the results have shown differences between the protein profiles that can interact with heme in uninfected and virus-infected cells.

These observations of heme-binding protein in-gel and on membranes raised questions about their binding properties in solution. UV-visible spectra of hemin (10 µM) diluted in PBS were recorded to analyze possible viral protein interactions with heme, revealing a Soret peak with λ max = 385 nm. Changes in hemin absorption spectra were observed after incubation with mock and virus-infected protein extracts. A redshift was noticed in both cellular extracts (λ max = 390 nm) (Figure 1c). Furthermore, mock and virus-infected cell extract (Figure 1d) increased the Soret band and protoporphyrin IX absorbance in both the Soret and Q bands.

### 2.2. SARS-CoV-2 Proteins Identification and Confirmation of Binding to Hemin and Hemoglobin

Shotgun proteomics was used to identify which SARS-CoV-2 proteins were interacting with heme. Initially, SARS-CoV-2-infected Vero cells total protein extract was subjected to SDS-PAGE, followed by excision of all visible bands and submission to processing for protein content identification by nanoelectrospray coupled to high-resolution tandem mass spectrometry (LC–MS/MS). This approach yielded virus protein identifications in the six bands indicated in Figure 2a. Nucleoprotein was identified in almost all excised bands (2, 3, 4, 5, and 6). Spike protein was identified in bands 1, 2, 3, and 6, while membrane protein was identified in bands 1 and 3. The non-structural proteins NSP3 and NSP2 were detected in bands 1 and 3, respectively (Figure 2a). The most abundant protein identified in band 4 was albumin, probably due to supplementation in the culture medium. Thus, to eliminate the excess of this contaminant protein, we initially depleted albumin by affinity chromatography. Next, the albumin-depleted protein extract of SARS-CoV-2 infected Vero cells was incubated with hemin-agarose beads to confirm that viral proteins can interact and bind heme. Finally, the hemin-agarose eluate containing the hemin-binding proteins was subjected to SDS-PAGE under denaturing and reducing conditions, revealing the presence of several viral proteins, as identified by LC–MS/MS (Figure 2b). In-band 7, nucleoprotein, spike, Nsp3, Nsp7, and membrane protein were placed. Nucleoprotein was also present in band 8 with the highest spectral counts and spike protein in lower abundance. The spectral counts and additional information for each band are shown in the Appendix A. Additionally, after the interaction, western blot with immunostaining using convalescent patient serum revealed two reactive bands at ~70 kDa and ~55 kDa in the total extract, one at ~70 kDa in the unbound fraction, and two bands (one intensely reactive at ~55 kDa and the other at ~45 kDa) in the bound fraction (Figure 2c).

The observation that viral proteins can bind to hemin suggested that this interaction could extend to Hb. Therefore, an overlay assay was performed with Hb on SARS-CoV-2 infected Vero cell extracts resolved by SDS-PAGE and transferred to NM (Figure 3a) to test this hypothesis. Haptoglobin, used as a positive control, displayed the predicted size for the β-chain at 40 kDa and showed that this approach could detect Hb binding proteins. Furthermore, it revealed two protein bands (~150 kDa and ~75 kDa) exclusively in SARS-CoV-2-infected cells while also found the other bands (~95 kDa, ~70 kDa, and ~50 kDa) in the mock cell extract.

### 2.3. Identification of the Hb Bind Motifs in the RBD and NTD of the S Protein

Based on the results from mass spectrometry (Figure 2a,b) and the overlay assay (Figure 3a), we set out to confirm that the spike glycoprotein interacts with Hb, performing a protein–protein interaction assay containing the RBD motif and NTD region based on peptide microarray (Spot Synthesis). The results are presented in Figure 3b (top panel). Six unique peptide sequences in RBD and eleven in NTD were identified as interacting with Hb (Figure 3b, lower panel). Further, a molecular docking assay indicated the binding of the spike protein with the alpha and beta Hb domains (Figure 3c), showing binding energy of −460 kcal/mol (Appendix A). The data showed that this interaction could be coordinated by four hot spot amino acid residues (Appendix A) from both proteins: spike (Tyr341, Tyr351, Phe347, Arg346, Tyr451, and Leu452) and Hb (His46, His51, Gln55, and Asp48) (Figure 3d).

### 2.4. Identification of Hb, Heme, and PpIX Binding Motifs in the Nucleoprotein by In Silico Analysis

The analysis of 68 crystallized hemoglobin/heme complex structures (Appendix A) identified conserved amino acid residues (~97% in the same position) interacting with the heme group. Furthermore, His displayed the highest binding frequency to heme, while Ala had the lowest contribution (Figure 4a). From the mapped motifs in human Hb (Appendix A) three equivalent motifs in SARS-CoV-2 N protein—presenting different E-values (Appendix A): motif 1 = 3.4 × 10^−505^, motif 2 = 4.4 × 10^−532^ and motif 3 = 1.8 × 10^−628^—were identified (Figure 4b). These analyses also indicate a correspondence regarding the residues Tyr (in motif 1), Lys (in motif 2), and Leu and Phe (in motif 3) in heme-binding motif composition between human Hb and the Nucleoprotein of SARS-CoV-2 (Figure 4b).

The molecular docking assay was essential to investigate possible binding pockets in the SARS-CoV-2 Nucleoprotein. A cavity with a −9.8 kcal/mol binding energy and a 121 Å size was selected to be assayed. In this cavity, it was possible to identify the bonds of PpIX with motifs 1 and 3 (Figure 4c), with a predominance of hydrophobic bonds followed by polar bonds and hydrogen bonds (Figure 4d) with porphyrin propionate groups. Both motifs’ potential to contribute to the coordination of PpIX binding was related with Arg259 of motif 1 performing hydrogen bonding and Tyr333, Thr334, and Gly335 of motif 3 performing hydrophobic and polar bonds, respectively (Figure 4e).

### 2.5. Effect of the Hemoglobin, Hemin, and PpIX on the Adsorption and Viral Replication

The intense interaction between peptides in the RBD of the spike protein and the localization of Hb in the molecular model suggested that the presence of free Hb could competitively interfere with the binding of SARS-CoV-2 to ACE2 receptor on host cells. As such, we hypothesized that the addition of Hb could impair virus replication in an in vitro assay. Vero cells were pretreated with a sub-optimal Hb concentration (1 µM) for 1 h at 37 °C before their exposure to the virus and then infected with SARS-CoV-2 at MOI of 0.01 to test this hypothesis. Alternatively, the virus was preincubated with 1 µM of Hb for 1 h before adding Vero cells for another hour at 37 °C. After 24 h, culture supernatants were collected, and the production of infectious virus particles was quantified by plaque assay. The pretreatment of Vero cells with Hb did not affect virus production (Figure 5a, left column); however, the virus’s preincubation with Hb reduced approximately 52.67 ± 0.77% of virus replication. Convalescent plasma was a positive control for viral neutralization and inhibited viral replication (99.96% ± 0.03%) (Figure 5b, right column). Hemin and PpIX were also tested to evaluate if porphyrin-protein interactions could affect virus replication. Hemin displayed no or little effect in virus replication when used as a pretreatment of Vero cells or SARS-CoV-2, respectively (Figure 5a,b, center columns). PpIX inhibited SARS-CoV-2 replication by 87.32 ± 1.2 and 100% when used as a pretreatment of the cells or virus, respectively (Figure 5a,b, right columns).

Next, we analyzed if viral replication reduction was related to Hb binding interference with a receptor-mediated virus attachment. An adsorption inhibition assay was performed. The incubation of the virus with cells was performed at 4 °C for 1 h to minimize internalization, and then viral RNA was purified for quantification by RT-PCR. When SARS-CoV-2 was pre-incubated with Hb (1 µM) for 1 h at 37 °C and then applied to Vero cells at an MOI of 0.01, a reduction of 59.7 ± 20.9% was measured in the virus adsorption (Figure 5c). Hemin (1 µM) reduced virus attachment by approximately 40%, and PpIX reduced virus attachment by 61.3 ± 18.0%. When cells were treated with 1 µM of either Hb, hemin, or PpIX, after the initiation of infection by SARS-CoV-2, virus replication was inhibited 99.5 ± 0.4% by Hb, 100% by PpIX, and 29.1 ± 7.2% by hemin (Figure 5d).

## 3. Discussion

Severe COVID-19, caused by SARS-CoV-2, typically leads to pneumonia and acute respiratory distress syndrome (ARDS). However, the growing list of evidence indicates a systemic impairment that leads to multiorgan failure. During an infection, an imbalance in the immunological response can produce a “cytokine storm” and numerous other pathophysiological processes such as hypoxemia, thrombosis, pulmonary embolism, encephalopathy, myocardial injury, heart failure, and acute kidney injury [21]. Hematological dysfunction in severe COVID-19 includes low levels of erythrocytes and an increased variation in the red blood cell distribution width (RDW) [8,9]. Recent reports on the immune effects of COVID-19 have highlighted immune thrombocytopenia and autoimmune hemolytic anemia [22]. Other evidence suggests an increase in hemophagocytosis related to elevated ferritin levels in COVID-19 [23]. In hemolytic disorders, the release of high levels of Hb and heme triggers the pro-inflammatory response, complement activation, and procoagulant and pro-oxidative environment [24,25].

Recently, an in silico analysis hypothesized that viral proteins could bind Hb beta-chains that would be expected to interfere with O_2_ transport and heme metabolism [6]. Another finding suggested that the sequence similarity between SARS-CoV-2 spike protein and hepcidin, a peptide hormone involved in iron metabolism, could lead to an imbalance of iron metabolism [7]. Yet, in the absence of experimental data, it is not clear if SARS-CoV-2 proteins actually can interact with Hb and displace iron from heme. 

To decipher these possible interactions, in silico analysis was performed to identify heme-binding motifs in viral proteins. Commonly, heme coordination can occur by hydrogen bonds via propionate groups, π–π stacking, electrostatic and hydrophobic interactions. The critical heme-coordinating amino acids are histidine, cysteine, and tyrosine; methionine and lysine occur at a lower frequency [26,27]. Here, in silico analysis supported the potential interaction of the porphyrin ring with Tyr333 of the viral N protein in addition to hydrophobic and hydrogen bonds via propionate side chains. An open question is whether heme is a necessary endogenous cofactor recruited by viral proteins during replication and translation. The primary function of the N protein is to bind RNA. This protein comprises an N-terminal domain containing the RNA-binding site, a C-terminal dimerization domain, and a central linker region rich in serine and arginine [28]. The heme-binding motifs identified in the N protein suggest another function for this protein. The analyses of other N protein motifs identified a nuclear localization signal and a nuclear export signal that offer dynamic nuclear-cytoplasmic trafficking that can involve transcription regulation [28].

N protein binding to heme might have an important implication in viral and cellular transcriptional regulation since heme regulates multiple transcription factors in the nucleus, modulating the expression of various genes [29]. Interestingly, high-confidence protein–protein interactions between SARS-CoV-2 and human proteins identified N protein associated mainly with components of cellular translational machinery [30]. N protein was reported to interact with two subunits of casein kinase 2 (CK2), a protein involved in activation of heme regulator inhibitor (HRI) kinase that phosphorylates eukaryotic initiation factor 2 (eIF-2), inhibiting translation [31]. N protein could be involved in dissociation or recycling of heme from HRI, modulating the phosphorylation of eIF2-α [32]. Some viruses do not require eIF-2 and even induce host translation shutdown [33]; possibly, these events consist of a manipulation route for viral protein synthesis. 

Electrophoresis of mock and SARS-CoV-2 infected cells revealed different heme-binding profiles. The electrophoretic mobility in the TMB in-gel staining assay suggests that spike fragment (S1 and S2, ~70 kDa) and Nucleoprotein (55 kDa) could bind hemin. Moreover, heme-binding interactions in solutions were changed by viral infection. Due to their hydrophobicity, heme and PpIX have low solubility in an aqueous solution and tend to aggregate. Binding complex with proteins can cause redshift, increase the Soret band’s intensity, and change the spectra profile of the Q band [34]. However, these results could also indicate a viral modulation of cellular hemeproteins or heme-binding proteins.

Shotgun proteomics was performed to identify SARS-CoV-2 heme-binding proteins to refine the analysis. Identification of proteins in SARS-CoV-2-infected Vero cells total extract revealed structural and non-structural proteins. Spike holoprotein was placed in the expected gel migration range (Band 1), but also cleavage products S1 and S2 produced by proteases (TMPRSS2, furin, cathepsin) were observed at ~70 kDa bands (Band 2 and 3). Interestingly, the SARS-CoV mass spectrometry analysis also identified spike protein in bands of relative molecular masses divergent from their theoretical sizes, suggesting glycosylation and cleavage [35]. As expected, Nucleoprotein was frequently identified and found in bands at 55, 45, and 40 kDa. The 55 kDa band in the TMB in-gel staining assay firmly retained hemin. The N protein presence in bands with different predicted molecular masses has been previously noted in SARS-CoV but attributed to protein degradation [36].

Interestingly, N protein was identified in a 46 kDa band of nucleus fraction of infected cells [36]. The transition to a higher molecular mass has been observed for nucleoprotein protein using mass spectrometry [35]. It can be indicative of the complexity of protein–protein binding dynamics in the replication of SARS-CoV-2. Since albumin in culture medium masks low abundance proteins identification [37], we performed a serum depletion followed by affinity chromatography using hemin-agarose beads. This approach increased the identification of proteins, and after purification, the N protein was the most abundant protein identified by mass spectrometry in the 45 kDa band. A variety of protein–protein interactions are essential for SARS-CoV-2 replication and viral assembly. Such is the case for the virus structural proteins, which have been reported to interact with each other [38]. In the hemin-agarose eluate, the M protein was identified in band 7 at a higher relative molecular mass position than its predicted molecular mass of 25 kDa. It is a transmembrane glycoprotein that comigrate with the other structural proteins-spike and N. While the interaction of the M protein with the N protein has been observed previously and appears to be necessary to viral assembly [39], this comigration was not scanned in the total extract of proteins from virally infected Vero cells before the inclusion of the heme-agarose beads. N and S proteins are highly immunogenic, unlike M proteins [40]. Spike protein appears to bind weakly to hemin beads since no 70 kDa noticed immunoreactive bands in the eluate fraction.

The heme-binding capacity of viral proteins leads us to question if this could extend to Hb. In the overlay assay, approximately 150 kDa and 70 kDa bands were exclusively found in the virus-infected cell extract, matching the spike holoprotein’s predicted size and fragments S1 and S2. Identifying spike protein as a significant viral protein in these bands lead us to investigate if this protein could interact, through its RBD region, with Hb. The spot-synthesis analysis demonstrated several areas of the RBD of the spike protein bind Hb, corroborating the molecular docking analysis. The apparent strength of binding according to the strong signals and Hb’s positioning on the spike protein suggests that its interaction with the spike could hinder viral entry and subsequent replication. One of the highest spot signals (448YNYLYRLFRKSNLKP463) was observed within the receptor-binding motif region (RBM; 437–508). This region is responsible for binding to ACE2 and includes hot spot residues that bind Hb (Tyr451 and Leu452). It is essential to highlight that, so far, the most frequent mutations in the RBD region of new variants are located in RBM (Tyr453, Gly476, Phe486, Thr500, and Asn501); a strong signal above 90% was noticed in the sequence (494SYGFQPTNGVGYQPYRVVVL513), which contained two frequent sites of mutations [41]. Moreover, several peptide sequences were identified in the S1-NTD domain, growing evidence suggests that initial contact and interaction of the virus to the epithelium is mediated by sialic acid, other glycans, and NTD [42].

Additionally, NTD has been a target for antibody neutralization; mAb bind prevents conformational changes in S protein and blocks membrane fusion [43]. Interestingly, the conserved 72GTNGTKR78 motif, associated with protein and sugar receptor binding [44], was observed to interact with Hb and peptide sequence 66HAIHVSGTNGTKRFD80. Collectively, the screening of a small library of linear peptides contributes with evidence of several sites of interaction between Hb and spike S1 that could interfere with viral entry. However, some protein–protein interactions and protein-receptor binding may depend on the tertiary structure and conformation of the protein. Therefore, to clarify if this interaction could occur in a complex cellular environment, in vitro assays were performed.

The ability of porphyrins to interfere in receptor binding was observed through in vitro assays. Hemin alone had little effect on viral attachment and replication, suggesting that hemin has little to no direct interactions with the spike protein or ACE-2. As intracellular concentrations of hemin are highly regulated, and hemin can be exported or degraded by heme-oxygenase 1, the absence of an effect is consistent with its physiology. In contrast, exposure of either cell or virus to PpIX dramatically reduced viral load. Similar outcomes for PpIX and verteporfin have been recognized against SARS-CoV-2 since treatment with porphyrins interferes with the ACE2 and spike that would impair viral entry. Moreover, these drugs were able to inhibit viral RNA production, suggesting other potential mechanisms of action [19]. Likewise, our results showed that PpIX at 1 µM could inhibit viral replication after 24 h. PpIX is hydrophobic and could interact with membranes; thus, porphyrin interaction with the viral envelope can induce destabilization and oxidation [45].

Pretreatment of viral particles with Hb reduced approximately 50% of both viral replication and adsorption, demonstrating spike/RBD-ACE-2 fusion impairment. Furthermore, Hb treatment’s effect after viral infection was higher, reducing 99% of viral replication. Modulation of viral replication by extracellular free Hb can occur via prooxidant activity, direct interactions of globin or heme to cell components and signaling pathways, and induction of heme oxygenase [46]. Downregulating HO-1 is a strategy for the optimization of virus replication and to evade host antiviral mechanisms for hepatitis C virus (HCV), hepatitis B virus (HBV), and pseudorabies virus (PRV) infection [47,48,49]. HO-1 is a critical stress-induced enzyme that promotes antioxidant, antiapoptotic, and anti-inflammatory activities via downstream metabolites, such as biliverdin and bilirubin [50]. Biliverdin impairs HCV replication by inducing an interferon response [51]. Induction or overexpression of HO-1 inhibits some viruses like influenza [51], human respiratory syncytial virus (RSV) [52], and Zika [53].

Additionally, binding of spike with Hb may have a role in COVID-19 pathophysiology. Meta-analysis revealed that severe COVID-19 patients had decreased hemoglobin levels, lower RBC count, and higher RDW than moderate COVID-19 cases [54]. Although a combination of events plays a role in COVID-19 hypoxemia [55], the decrease in Hb levels contributes to hypoxia and is related to complications, ultimately leading to multiorgan failure [54]. Low Hb levels were attributed to hemolytic anemia driven by inflammation and iron dysmetabolism, interfering with erythropoiesis [22,54]; our results demonstrate that viral particles’ binding can also be implicated. In the scenario of hemolytic anemia, an excess of heme/Hb increases ROS levels and tissue damage leading to vascular injury and ferritin overexpression [56]. High ferritin levels are a hallmark of COVID-19, eventually contributing directly to inflammation and lung injury since ferritin is pro-inflammatory and leads to ferroptosis [7,57]. A recent report found that even after two months from the onset of disease, 30% of patients still presented iron deficiency, hyperferritinemia (38%), and anemia (9.2%) correlated with disease severity [57].

Additionally, free heme and hemoglobin are involved in hemostasis and thrombosis. Hb enhances platelet activation by scavenging of nitric oxide [24] and induces platelet aggregation contributing to prothrombotic events [58]. The assembly of viral particles and Hb capping of spike could contribute to COVID-19 thromboembolic events and aggravate lung injury in critically ill patients. Hemolysis in the intravascular and alveolar spaces results in Hb release that can contribute to organ dysfunction [59]; in ARDS, it is proposed that Hb-mediated damage by cell surface receptor binding on the alveolar epithelium is independent of oxidative stress [60].

Overall, our demonstration that SARS-CoV-2 proteins can bind to heme or Hb may have clinical implications. Mainly, Hb’s interaction with spike opens new therapeutic perspectives due to significant virus attachment and replication inhibition. Moreover, this binding could potentially increase or drive hematological disorders and thrombosis observed in severe COVID-19. Although there are still knowledge gaps on viral-host cell complex interplay and disease pathophysiology, the data presented here will contribute to scientific discussion. The interactions observed using different methods were performed in vitro and in silico and more research will be needed to confirm the relevant implications of these heme-protein interactions and correlations with COVID-19 physiopathology in vivo.

## 4. Materials and Methods

### 4.1. Motifs Identification

The MEME Suite server identified the motifs for binding to the heme group on the SARS-CoV-2 nucleoprotein [61]. First, protein sequences were mapped using the server Sequence Annotated by Structure [62] from 68 crystallized structures with a heme group. Then, 36 Hb sequences deposited in the UniProt database (www.uniprot.org, accessed on 8 September 2020) were used against 13 nucleoprotein sequences from SARS-CoV-2.

### 4.2. Molecular Docking

#### 4.2.1. Receptor-Ligand

Molecular docking assays were employed to predict the binding modes of the Sars-CoV-2 nucleoproteins (PDB code—6zco; https://www.rcsb.org/, accessed on 8 September 2020) and ligand heme prosthetic group (Fe-protoporphyrin) complexes by using the DockThor server (https://dockthor.lncc.br/v2/, accessed on 8 September 2020). Structures with positional root mean square deviation (RMSD) ≤ 2 Å were clustered and selected with the most favorable free energy of binding.

#### 4.2.2. Protein–Protein Interaction

This assay was performed with human hemoglobin (PDB code—4x0i) and spike protein (PDB code—7kn5), by using the RosettaDock server (http://rosettadock.graylab.jhu.edu, accessed on 20 September 2020). In addition, the alpha and beta chains of hemoglobin were assessed in the protein interaction assays and evaluated using the HotRegion database (http://prism.ccbb.ku.edu.tr/hotregion/index.php, accessed on 20 September 2020).

### 4.3. Cell Culture, Virus Expansion, and Virus Tittering

African green monkey kidney (Vero, subtype E6, ATCC^®^CRL-1586™) cells were cultured in media consisting of high glucose DMEM complemented with 10% fetal bovine serum (FBS; HyClone, Logan, UT, USA), 100 U/mL penicillin, and 100 μg/mL streptomycin (Pen/Strep; Thermo Fisher Scientific, Waltham, MA, USA). Cells were maintained in a humidified atmosphere with 5% CO_2_ at 37 °C. The SARS-CoV-2 used in these studies (GenBank #MT710714) was isolated from a nasopharyngeal swab obtained from a consenting patient with COVID-19, as confirmed by RT-PCR. The virus was expanded in Vero E6 cells at a multiplicity of infection (MOI) of 0.01, according to WHO guidelines that mandate all procedures related to virus cultures be performed in a biosafety level 3 (BSL3) facility. 

Virus titer was defined as plaque-forming units (PFU)/mL. Briefly, Vero E6 cells were seeded into 96-well plates at 2 × 10^4^ cells/well for 24 h before exposure to a serial dilution of expanded SARS-CoV-2 for 1 h at 37 °C. Next, a semi-solid high glucose DMEM medium containing 2% FSB and 2.4% carboxymethylcellulose and cultures were incubated for 3 days at 37 °C. Then, the cells were fixed with 10% formalin for 2 h at room temperature. Finally, the cell monolayer was stained with 0.04% solution of crystal violet in 20% ethanol for 1h. Virus stocks were stored at −80 °C until use.

### 4.4. Yield Reduction Assays and Virus Titration

Vero cells were seeded into 96-well plates at a density of 2 × 10^4^ cells/well for 24 h at 37 °C before exposure to SARS-CoV-2 at an MOI of 0.01. After a 1 h incubation, the inoculum was removed, and cells were incubated in a medium containing 1 μM of the experimental compounds diluted in DMEM with 2% FBS. Alternatively, two experimental conditions were performed: (i) preincubation of the virus with the compounds (1 μM) for 1 h at 37 °C before their addition to Vero E6 cells (MOI of 0.01) for an additional hour, or (ii) preincubation of Vero cells with the compounds (1 μM) for 1 h at 37 °C before their exposure to the virus (MOI 0.01). After 24 h, supernatants were collected for virus titration (PFU/mL) as described in the previous section.

### 4.5. Adsorption Inhibition Assays 

The virus was incubated with a compound (1 µM) for 1 h and then added to monolayers of Vero E6 cells in 48-well plates (5 × 10^5^ cells/well) at an MOI of 0.01 for 1 h at 4 °C. The medium with the virus was removed, and cells were washed three times with medium, before lysis buffer addition. Total viral RNA was extracted using QIAamp Viral RNA (Qiagen, São Paulo, SP, Brazil), according to the manufacturer’s instructions. Quantitative RT-PCR was performed using GoTaq^®^ Probe qPCR and RT-qPCR Systems (Promega, Madison, Wisconsin, EUA) in a StepOne™ Real-Time PCR System (Thermo Fisher, Waltham, MA, USA). Amplifications were performed as 25 µL reactions containing 1× reaction mix buffer, 50 µM of each primer, 10 µM of the probe, and 5 µL of RNA template. Primers, probes, and cycling conditions followed the recommendations of the Centers for Disease Control and Prevention (CDC) protocol for the detection of the SARS-CoV-2 [63]. In addition, a virus quantification standard curve was included [64]. 

### 4.6. SDS-PAGE and Hemin-Binding Blots

Vero E6 cells were infected with SARS-CoV-2 at a multiplicity of infection (MOI) of 0.01 for 1 h at 37 °C. The inoculum was removed and added to a fresh culture medium. Protein extracts (20 µg) were obtained 24 h post-infection by lysing the cell monolayer with lysis buffer (100 mM Tris-HCl pH 8.0, 150 mM NaCl, 10% glycerol, 0.6% Triton X100). Protein extracts were resolved by SDS-PAGE (10%) using the Laemmli buffer [65]. For hemin-binding blots, proteins were transferred to NM and then rinsed with Tris-buffered saline (TBS; 10 mM Tris-HCl pH 8.0 containing 150 mM NaCl) plus 0.1% Tween 20 (TBST) followed by 1 h incubation with TBS containing hemin (2 µM). Membranes were subsequently washed three times for 30 min with TBST and revealed with a solution containing 0.1 mg/mL 3,3′ diaminobenzidine (DAB), 0.1% H_2_O_2_, 10 mM HEPES, pH 6.2, and 100 µM CaCl_2_ overnight at 4 °C in the dark [66]. Alternatively, heme-binding proteins were evaluated by 1 h room temperature incubation of the protein extracts (20 µg) with hemin (300 µM) in 250 mM Tris-HCl, pH 8.0, 5 mM EDTA, and 10% glycerol, followed by SDS-PAGE [67]. Then, gels were washed for 1 h with PBS containing Triton X-100 (2.5%) and equilibrated for 30 min with sodium acetate 0.5 M (pH 5.0). Heme binding proteins were revealed with 2 mg/mL 3,3′,5,5′-tetramethylbenzidine (TMB) dissolved in 15 mL of methanol and 35 mL of 0.5 M sodium acetate (pH 5.0). Then, 300 μL of 30% H_2_O_2_ solution was added, and the reaction was carried out for 30 min in the dark. After a blue-colored band developing, indicative of a heme-protein complex, the gels were washed with sodium acetate (pH 5.0) and isopropanol (30%) solution. Protein extract without preincubation with hemin was used as a negative control.

### 4.7. Hemin and PPIX Binding Assay

Hemin chloride (10 μM) diluted in NaOH (0.1 M) was added to a quartz cuvette containing 20 μg of protein extract from Vero cells and SARS-CoV-2 infected Vero cells, and the spectrophotometric analysis was carried out at 300–700 nm (SpectraMaxM2e, Molecular Devices, CA, USA). For protoporphyrin IX (PPIX) binding assays, 20 μg of each protein extract was incubated with PPIX solution (1 mM in DMSO followed by dilution to 1 µM in phosphate buffer saline pH 7.4) and submitted to absorbance spectral analysis.

### 4.8. Protein Purification and Hemin-Agarose Binding Assay

Protein extract, obtained from infected Vero E6 monolayers 24 h post-infection lysed with 100 mM Tris-HCl pH 8.0, 150 mM NaCl, 10% glycerol, 0.6% Triton X-100, was initially subjected to affinity chromatography using a protein-A/anti-BSA mAB matrix (Sigma-Aldrich; B2901) to remove excess albumin (a heme-binding protein contaminant). Then, hemin-agarose was used to isolate heme-binding proteins. Then, hemin-agarose was used to isolate heme-binding proteins. Briefly, 200 mL of hemin-agarose (Sigma-Aldrich) was washed three times in 1 mL of 100 mM NaCl, 25 mM Tris-HCl (pH 7.4) for 5 min, and centrifugated at 700× *g*. Next, hemin-agarose was incubated for 1 h at 37 °C, under agitation, with SARS-CoV-2 protein extracts (800 µg). Next, the unbound proteins were removed by washing three times with equilibration buffer, and beads incubated for 2 min with elution solution (2%, SDS, 1% β-mercaptoethanol in 500 mM Tris HCl, pH 6.8) followed by boiling at 100 °C for 5 min [68]. Finally, total extract, unbound (supernatant) fraction, washing fraction, and hemin-agarose bound proteins were resolved by SDS-PAGE (10%) and stained with Coomassie Blue R-250 or transferred to an NM subsequently blocked with TBS-T (Tris-buffer saline, 0.1% Tween 20, pH 7.5) and 5% defatted milk. Immunostaining was later performed by incubating membranes overnight at 4 °C with a pool (*n* = 10) of COVID-19 convalescent serum (1:200). Then, membranes were washed and incubated with HRP-conjugated anti-human IgG antibody (Sigma-Merck; 1:10,000) followed by chemiluminescence detection.

### 4.9. Overlay Assay

Protein extract (20 µg), obtained as described above, was resolved by SDS-PAGE (10%) and transferred to an NM (Bio-Rad 2 µm). After blocking (2% BSA in TBS-T buffer) of the free sites, the membrane was incubated (overnight at 4 °C) under agitation with a solution of 10 µg/mL of human adult hemoglobin (Sigma-Aldrich, Burlington, MA, USA), followed by a new incubation with anti-Hb antibody (1:5.000; Sigma-Aldrich, H4890). The antigen-antibody complex was revealed by chemiluminescence. Haptoglobin (Sigma-Merck, Burlington, MA, USA) was used as a control. 

### 4.10. Spot-Synthesis 

The sequence of the spike protein (P0DTC2) receptor-binding domain and N-terminal domain (NTD) was retrieved from UniProt. Human alpha-S1-casein (P47710) was used as negative control and human haptoglobin (P00738-2) as a positive control. A library of 15 amino acid peptides with a 5-residues overlap was designed to represent the entire coding region of RBD (aa 319–541), NTD (aa 1–303), and partial sequence of haptoglobin (121–127; 226–234; 253–267; 283–289; 318–327) and alpha-S1-casein (31–40). Peptides were synthesized onto cellulose membranes using an Auto-Spot Robot ASP-222 (Intavis Bioanalytical Instruments, AG, Köln, Germany) [69,70]. Briefly, membranes containing the synthetic peptides were washed with TBS-T and then blocked with TBS-T containing 1.5% BSA under agitation for 2 h at room temperature. After extensive washing with TBS-T (Tris-buffer saline, 0.1% Tween 20, pH 7.0), membranes were incubated overnight with Hb (5 µg/mL) dissolved in TBST + BSA (0.75%). After incubation, membranes were washed with TBS-T, followed by additional incubation with anti-human Hb antibody produced in rabbit (Sigma Aldrich, H4890) for 90 min. Subsequently, the membrane was washed with TBS-T and incubated for 90 min with anti-rabbit IgG antibody conjugated to alkaline phosphatase (Sigma-Aldrich), diluted 1:5000 in TBS-T solution containing 0.75% BSA. Washes were performed with TBS-T followed by the addition of substrate for chemiluminescent alkaline phosphatase Tropix^®^. Next, membranes were washed three times with TBS-T, and then the buffer was exchanged to CBS (50 mM citrate-buffer saline) before the addition of the chemiluminescent enhancer Nitro-Block II. The chemiluminescent substrate Super Signal R West Pico was applied and immediately detected the signals generating a digital image file. The signal intensities were quantified using TotalLab (v 2009, Nonlinear Dynamics, Newcastle-Upon-Tyne, UK) software [67]. Binding motifs were considered analyzing the spots with signal intensity greater than or equal to 50% of the highest signal.

### 4.11. In-Gel Trypsin Digestion of Proteins

Protein spots were excised from gels using sterile stainless steel scalpels, transferred to 0.5 mL tubes, and cut into smaller pieces. In-gel digestion with trypsin (Promega V511A) was performed according to [71] with some modifications [72]. The addition of 100 μL of 65 mM dithiothreitol (DTT) for 30 min at room temperature was followed by alkylation with 100 μL of a 200 mM iodoacetamide solution 30 min (in the darkroom). After washes and trypsinization, the final 80 μL peptide-containing samples were concentrated by vacuum centrifugation to approximately 20 μL and stored at −20 °C until mass spectrometric analysis. Gel pieces from a “blank” region and the bovine sera albumin (BSA) molecular mass marker were negative and positive controls.

### 4.12. Identification of Proteins by Mass Spectrometry

The tryptic digests were analyzed in three technical replicates by reversed-phase nanochromatography coupled to high-resolution nanoelectrospray ionization mass spectrometry. Chromatography was performed using a Dionex Ultimate 3000 RSLCnano system coupled to the HF-X Orbitrap mass spectrometer (Thermo Fischer, Waltham, MA, USA). Samples (4 µL per run) were initially applied to a 2 cm guard column, followed by fractionation on a 25.5 cm PicoFrit^TM^ Self-Pack column (New Objective, Inc., Woburn, MA, USA) packed with 1.9 μm silica, ReproSil-Pur 120 Å C18-AQ (Dr. Maisch/Germany). Samples were loaded in 0.1% (*v*/*v*) formic acid (FA) and 2% acetonitrile (ACN) onto the trap column at 2 μL/min, while chromatographic separation occurred at 200 nL/min. Mobile phase A consisted of 0.1% (*v*/*v*) FA in water, while mobile phase B consisted of 0.1% (*v*/*v*) FA in ACN. Peptides were eluted with a linear gradient from 2 to 40% eluent B over 32 min, followed by up to 80% B in 4 min. Lens voltage was set to 60 V. Full scan MS mode was acquired with a resolution of 60,000 (FWHM for *m/z* 200 and AGC set to 3 × 10^6^). Up to 20 most abundant precursor ions from each scan (*m/z* 350–1400) were sequentially subjected to fragmentation by HCD. Fragment ions were analyzed at a resolution of 15,000 using an AGC set to 1 × 10^5^. Data were acquired using Xcalibur software (version 4.2.47).

### 4.13. Peptide Identification and Protein Inference

All MS/MS spectra were analyzed using PEAKS Studio X Plus (Bioinformatics Solutions, Waterloo, Canada). First, peptide identification was performed against *Chlorocebus sabaeus* reference proteome at the UNIPROT database under ID UP000029965, plus the SARS-CoV-2 reference proteome at the same database UP000464024 (downloaded 3 July 2020). Data refinement applied the precursor correction (mass only). Next, PEAKS de novo analysis was run assuming trypsin digestion, with a fragment ion mass tolerance of 0.02 Da and a parent ion tolerance of 10 ppm. Cysteine carbamidomethylation (+57.02 Da) was set as fixed modification; PEAKS DB analysis was performed using these same parameters, plus the possibility of up to two missed enzyme cleavages and nonspecific cleavage at both sides of the peptides. Finally, post-translational and other possible modifications were searched using the PEAKS PTM algorithm—with the same parameters described above and a maximum of two variable shifts per peptide was allowed—against a protein subdatabase composed only by protein entries found by the previous PEAKS De Novo and PEAKS DB searches. PEAKS decoy fusion approach estimated false discovery rates (FDR). A peptide-spectrum match FDR of 0.1% and protein identifications with at least two unique peptides were the criteria used to establish FDR values at peptide and protein levels smaller than 1%.

## 5. Conclusions

This study identified that the S, N, M, Nsp3, and Nsp7 proteins of the SARS-Cov2 could recruit Hb and its metabolites, heme, and protoporphyrin. Furthermore, we demonstrated that virus structural (N, S, and M) and non-structural (Nsp3 and Nsp7) proteins interacted with hemin. In addition, the spike protein RDB motifs and NTD domain (from SARS-CoV-2) bind Hb, and the sites of heme-binding for the N protein were disclosed. Finally, we demonstrated that—to different extents—Hb, hemin, and PpIX could block the virus adsorption and replication in vitro.

## Figures and Tables

**Figure 1 ijms-22-09035-f001:**
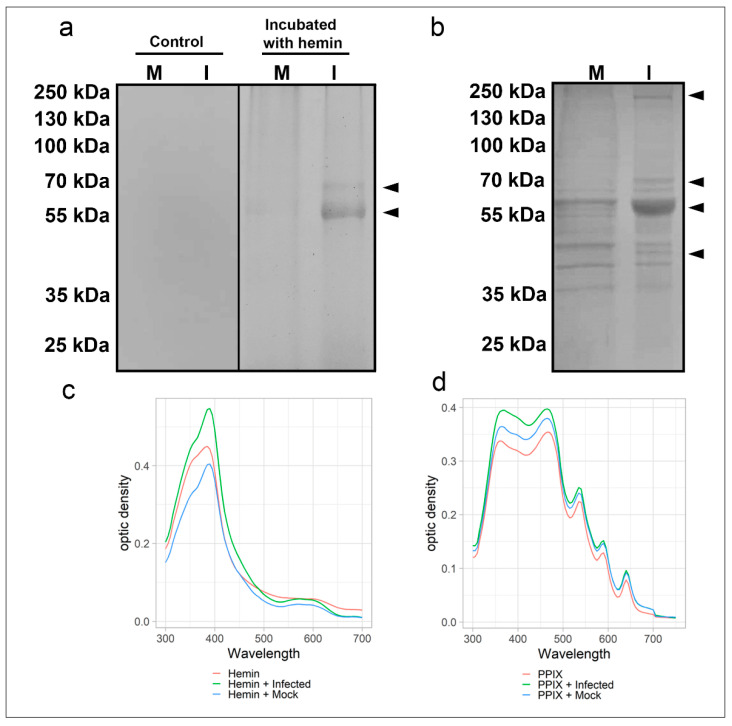
Heme-binding proteins in extracts from SARS-CoV-2 infected and non-infected Vero cells along with a spectroscopic binding analysis in solution. Total protein extracts (20 µg) from virus-infected (I) and mock-infected (M) were: (**a**) incubated with 300 µM of hemin for 1 h at 25 °C, resolved by SDS-PAGE, renatured and exposed to TMB in-gel to reveal hemin or (**b**) resolved by SDS-PAGE, transferred to NM and incubated with hemin (2 µM) for 1 h before revealing hemin-protein complexes by DAB. Arrowheads indicate bands of hemin-protein complexes. UV-visible spectra of hemin (**c**) alone (10 µM, red) or in total protein extracts (20 µg) of virus-infected Vero cells (green) and mock-infected cells (blue). UV-visible spectra of PpIX (**d**) alone (5 µM, red) or in total protein extracts (20 µg) of virus-infected Vero cells (green) and mock-infected cells (blue). Changes in the Soret peak and Q-bands are observable. Data are representative of two independent experiments. Source: data are provided as a source data file.

**Figure 2 ijms-22-09035-f002:**
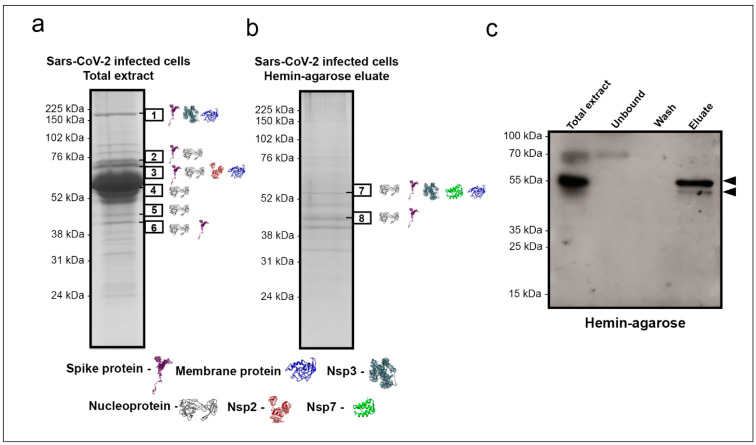
Heme affinity-purified proteins from total extracts of SARS-CoV-2 infected Vero cells and identification by mass spectrometry. SDS-PAGE resolved and Coomassie blue stained proteins: (**a**) whole virus-infected Vero cells extract and (**b**) eluate from a hemin-agarose column purification (after albumin depletion). Bands excised for protein identification by LC–MS/MS are indicated by black boxes 1–8. The proteins identified were N (P0DTC9), M (P0DTC5), S (P0DTC2), and replicase polyprotein 1a/1ab (P0DTC1/P0DTD1). The peptides in the replicase polyprotein 1a/1ab were identified as Nsp2, Nsp3, and Nsp7. The identified proteins in each band are shown as 3D structures retrieved from the I-TASSER site (https://zhanglab.ccmb.med.umich.edu/I-TASSER/, accessed on 20 September 2020). (**c**) Western blot revealed with COVID-19 patient convalescence sera pool (*n* = 5); arrowheads indicate two bands showed in the eluate.

**Figure 3 ijms-22-09035-f003:**
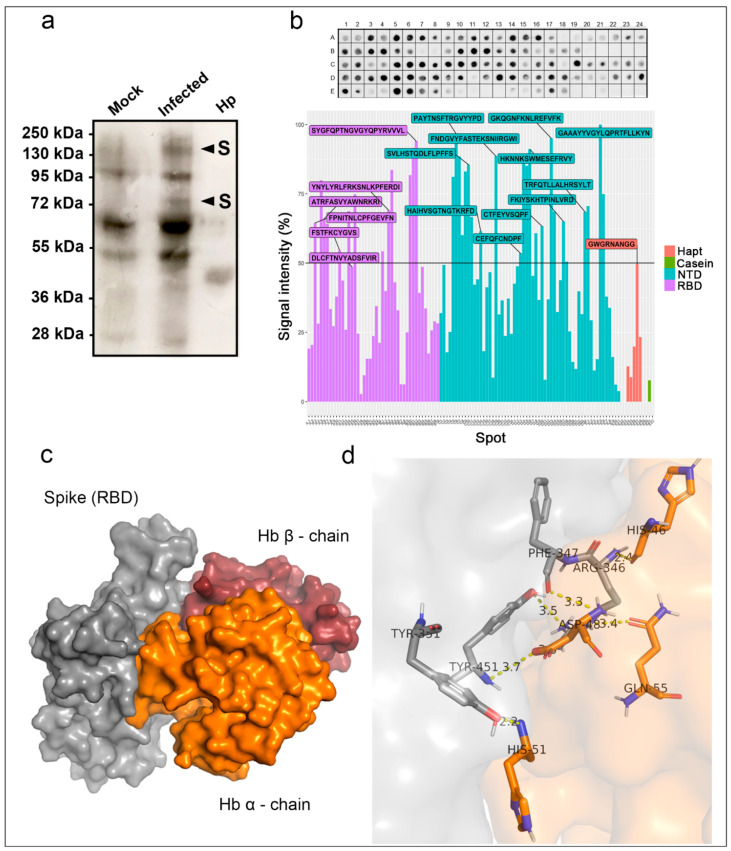
Hemoglobin (Hb) binding to spike protein and RBD enhancement. (**a**) Total protein extracts (20 µg) of SARS-CoV-2-infected Vero E6 cells (Infected) or mock-infected cells (Mock; 20 µg) were separated by SDS-PAGE (10%) along with haptoglobin (Hp; 2 µg), transferred to a nitrocellulose membrane, and serially incubated with Hb (10 µg/mL), anti-Hb antibody, and peroxidase-conjugated secondary antibodies. Two bands at 72 and 150 kDa were exclusively found in SARS-CoV-2 infected extract (arrowhead). (**b**) Spot synthesis analysis with a library of 15-mer peptides offset by five amino acids to represent the RBD region (purple) and NTD (blue) of the spike protein, haptoglobin (Hapt; red), and casein (green) as a positive and negative control, respectively. Sequences were synthesized directly onto a cellulose membrane followed by probing with Hb (5 μg/mL) and revealed by anti-human Hb antibodies. The top panel shows the chemiluminescent image of signals from peptides bound to Hb. The bottom panel shows a graph of the signal intensities normalized to the maximum signal. An intensity level above 50% defined Hb-reactive peptides. Molecular docking of SARS-CoV-2 spike protein with human hemoglobin. (**c**) Interaction of spike protein (gray) with α-chain Hb (orange) and β-chain (red). (**d**) Representation of amino acid residues binding to spike protein (sticks and gray) with α-chain of hemoglobin (orange).

**Figure 4 ijms-22-09035-f004:**
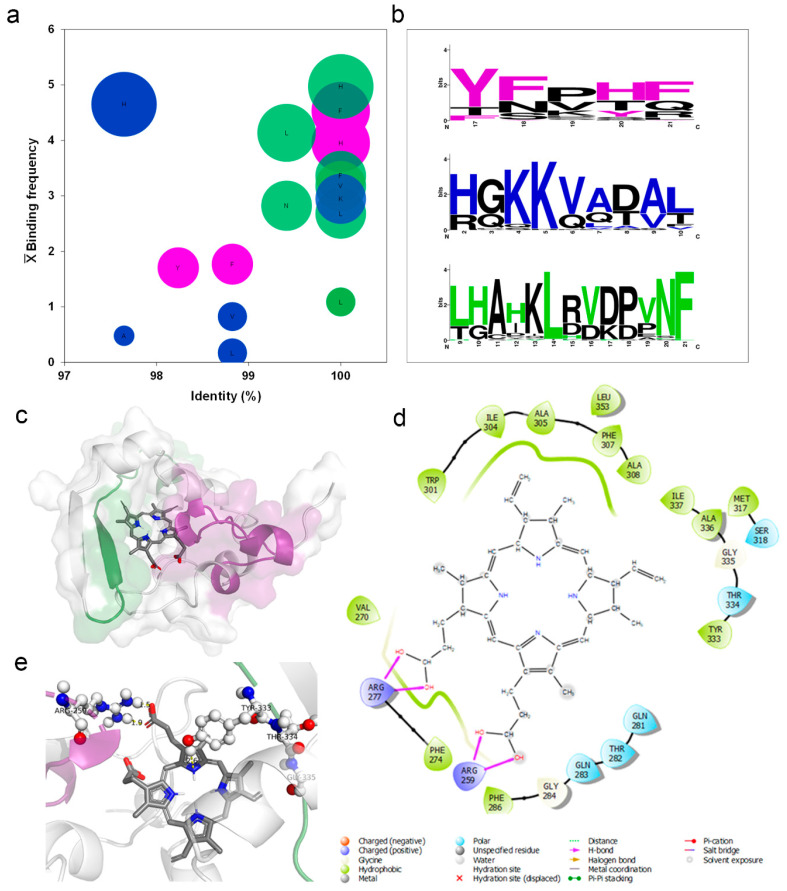
Heme-binding motifs were identified by an in silico analysis. (**a**) The heme-binding motifs for 68 Hb structures deposited in the PDB data bank were mapped to define the frequency of residues occurrence ($\bar{X}\;$Binding frequency) and degree of identity (%). The ball size is relative to the number of connections (5 to 1), displayed from biggest to smallest in size. (**b**) Three binding motifs were identified in the N protein of SARS-CoV-2 by an analysis in the MEME-Suite server of Hb amino acid sequence obtained from UniProt sever. Colored amino acids related to residues in Hb that interact with heme. Black residues form part of the motif without binding heme. The size of the amino acid relates to the number of occurrences (bits). Molecular docking of SARS CoV-2 N protein with human PpIX. (**c**) Binding position of PpIX (sticks) indicating the orientation of binding with N protein (surface) to predicted motifs: purple = motif one and green = motif 3. (**d**) Two-dimensional (2D) representation indicating types of bonds that occur in N protein pocket bounded with PpIX. (**e**) Three-dimensional (3D) model showing nucleoprotein amino acid residues (ball and sticks) that compose motifs 1 and 3 coordinating the binding with the PpIX (bars).

**Figure 5 ijms-22-09035-f005:**
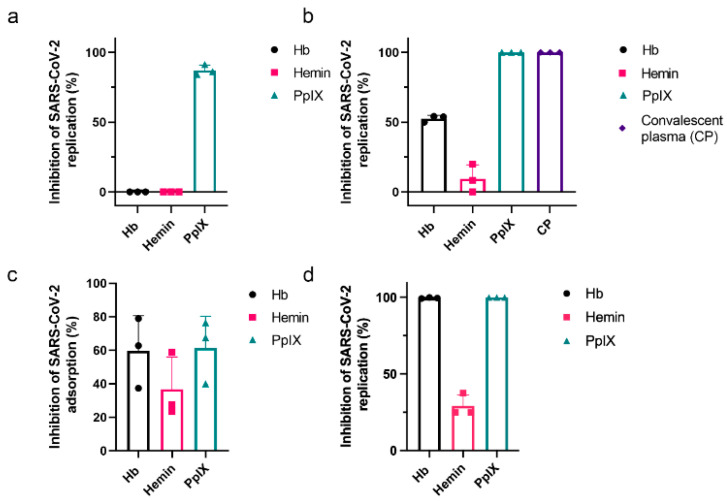
Inhibition of SARS-CoV-2 replication and attachment by porphyrins and Hb. Vero E6 cells (**a**) or SARS-CoV-2 (**b**) were pre-incubated with 1 µM of Hb, hemin, or PpIX for 1 h at 37 °C before the start of infection with an MOI of 0.01 for an additional hour at 37 °C. Culture supernatants were collected after 24 h, and the virus titer was determined by plaque assay (*n* = 3). A convalescent plasma of an infected patient (1:3 dilution) was used as the positive control. (**c**) SARS-CoV-2 was incubated at 1 µM Hb, hemin, or PpIX for 1 h at 37 °C before introducing Vero E6 cells at an MOI of 0.01 for 1 h at 4 °C. Rinsed cell monolayers were lysed, and RT-PCR quantified the virus content. Cumulative data (Hb, Hemin, PpIX: *n* = 3). (**d**) Vero E6 was infected with SARS-CoV-2 at an MOI of 0.01 for 1 h at 37 °C and then treated with 1 mM of hemin, PpIX, or Hb. After 24 h, supernatants were collected, and the virus titer was quantified by PFU/mL (Hb, Hemin, PpIX: *n* = 3). Data represented the percentage of inhibition compared to control (infected and untreated) and expressed as mean with standard deviation. Source data are provided as a Source Data file.

## Data Availability

The data presented in this study are available in Appendix A and on request from the corresponding author.

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
