# Peer review of "SARS-CoV-2 Proteins Bind to Hemoglobin and Its Metabolites"

_ijms, 2021, doi:10.3390/ijms22169035_

Round 1
Reviewer 1 Report
SARS-CoV-2 proteins bind to hemoglobin and its metabolites by Lechuga et al identifies the possible interaction of SARS-CoV-2 with hemoglobin. However, in absence of multiple experimental validations and missing vital controls as mentioned below undermine the present study. A major revision is required before the manuscript could be considered for publication.
Heme binding protein identification.
The author performed a hemin-binding assay using two different but theoretically similar approaches. Through both the approaches, the band pattern was strikingly different which suggests the lack of specificity of the SARS-CoV-2 protein in binding to hemin. However, in the mock run, there were no or few unrelated bands was identified. Did the author verify their findings with other alternative interaction studies/methods for example pull-down experiments etc., and support them with proper references?
In Section 2.4, the figure number would be 3 b, instead of 4?
In the Spot synthesis analysis, the authors did not use any positive or negative control, for example, ACE2 protein or unrelated protein that doesn’t bind to RBD of S protein. Or instead of SARS-CoV-2, MERS or CoV-1 peptide could have been used as a control? Similarly, the NTD region could also be included for spot assay, since growing evidence of mAb targeting NTD also neutralizes the virus and NTD and role with A, B, O blood group-related specificity been depicted in literature earlier?
Anti-Human Hb antibody detail is missing; please provide the details of all antibodies used in the text.
Line 406: action19??
Line 407: 24 h19??
The study lacks vital protein-protein interaction data that can support the claim and missing controls from the critical experiments jeopardize the result for the proposed hypothesis.
Author Response
SARS-CoV-2 proteins bind to hemoglobin, and its metabolites by Lechuga et al. identifies the possible interaction of SARS-CoV-2 with hemoglobin. However, the absence of multiple experimental validations and missing vital controls, as mentioned below, undermines the present study. Therefore, a major revision is required before the manuscript could be considered for publication.
Heme binding protein identification. The author performed a hemin-binding assay using two different but theoretically similar approaches. The banding pattern was strikingly different through both paths, which suggests the lack of specificity of the SARS-CoV-2 protein in binding to hemin. However, in the mock run, there were no or few unrelated bands was identified. Did the author verify their findings with other alternative interaction studies/methods, for example, pull-down experiments, etc., and support them with proper references?
R: We appreciate the work and effort of the reviewers who brought essential corrections to the text. Finally, we performed an affinity chromatography experiment using hemin beads. The use of this technique is commonly used to identify the heme-binding protein. After the assay, it is expected to find fewer bands with protein with high affinity with hemin. In the mock run, the low number of the unrelated band can be due to increased expression of viral protein or modulation and shift of cellular metabolism to replication and production of viral particles.
In Section 2.4, the figure number would be 3 b, instead of 4?
R: Changes was addressed to the manuscript.
In the Spot synthesis analysis, the authors did not use any positive or negative control, for example, ACE2 protein or unrelated protein that doesn't bind to RBD of S protein or, instead of SARS-CoV-2, MERS or CoV-1 peptide could have been used as a control? Similarly, the NTD region could also be included for spot assay, since growing evidence of mAb targeting NTD also neutralizes the virus and NTD and role with A, B, O blood group-related specificity been depicted in literature; earlier?
R: We appreciate the reviewer's suggestions. We performed a new Spot-synthesis experiment with a positive control using haptoglobin that bind hemoglobin peptide sequences and negative control with the casein, a commonly used blocking free sites. (Please see that in both experiments the free sites of the Spot synthesis membrane were previously blocked with BSA). In addition, we performed another experiment with RBD and the NTD. As expected, the Spot-synthesis results were reproducible, and despite slight differences in signal intensity (within different membranes analysed), peptide sequences in RBD that interacted with Hb were present in the second assay and identified sequences (Please see Source file). Also, several peptides in NTD strongly interacted with hemoglobin, more than the positive control peptides, reinforcing that these regions in Spike can bind Hb. This exciting result open new perspectives for the protein-protein interaction since Hb binding to NTD could also inhibit viral entry; some works reported interactions of NTD with sialic acid in the cell surface. The discussion now brings additional pieces of information.
Anti-Human Hb antibody detail is missing; please provide the details of all antibodies used in the text.
R: Changes were addressed to the manuscript introduced pertinent details in the text.
Line 406: action19??
R: Changes was addressed to the manuscript
Line 407: 24 h19??
R: Changes was addressed to the manuscript
The study lacks vital protein-protein interaction data that can support the claim, and missing controls from the critical experiments jeopardize the result for the proposed hypothesis.
R: We understand the referee's concern and performed new experiments with the controls of Spot-synthesis, and the assay showed several regions of Spike protein that bind Hb. We also used several different approaches: overlay assay, hemin affinity binding, mass spectrometry, in vitro, and in silico analysis to demonstrate that viral proteins can bind both hemin and Hb.

Reviewer 2 Report
Within this manuscript the authors demonstrate that structural (nucleoprotein, spike, membrane protein) and non-structural (Nsp3 and Nsp7) proteins of COVID19 interact with hemin and that preincubation of viral cultures with free hemoglobin ore PPIX might inhibit viral replication.
I am missing a section on strengths and weaknesses of the study, how reliable is the computational modelling, what might be different reasons for changes in the viral binding properties? What about pH changes due to the different components, indirect effects on cytokine expression within the cell culture etc.? How will tertiary protein structures and folding characteristics be affected by different compounds? Is the low number of independent experiments truly representative? Is in vitro and in silico comparable to the human in vivo situation?
Abstract: For me, referring to the high economic burden of COVID19 does not fit to the content of the article. I suggest to start with the hematological effects of COVID which might be explained by your work.
Methods: As far as I understand, Vero cells are isolated from kidney epithelial cells extracted from a monkey. Why have you chosen this kind of cells? Please discuss different cellular approaches in COVID research.
Discussion: Would it be possible to structure the discussion a bit more referring to the different results step by step?
Line 77: You seem to define protoporphyrin IX (PPIX) as heme, but I thought that only the chelation of PPIX with iron forms heme (iron PPIX). For this conversion the mitochondrial enzyme ferrochelatase is needed, please clarify.
Line 37/38 first sentence “in addition” second sentence “additionally”, perhaps it could be rewritten without redundancy
4.7. Why have you chosen 10uM heme to be added to 20ug of protein extract?
101/102: Why have you chosen the preincubation with 300 μM hemin?
Figure 3: How often has the experiment been repeated, why are you sure that there is no unspecific binding? There is a third line with Hp showing at least one band, could you specify?
Line 288: Please specify whether you used human adult hemoglobin and where it was isolated from (or purchased)
Fig. 5: What is the reason for performing only n=2 in the case of Hb-inhibition experiments?
Line 78, 82, 342, 344, 345, 350, 457, 474 The citations appear in a different style without brackets
469: Do you refer to hemoxygenase 1 (it is written 11)?
Line 564: I guess it should be 5x 105
Figure 5 displays a standard deviation based on two values. Isn’t this an overestimation of results?

Author Response
Within this manuscript, the authors demonstrate that structural (nucleoprotein, spike, membrane protein) and non-structural (Nsp3 and Nsp7) proteins of COVID19 interact with hemin and that preincubation of viral cultures with free hemoglobin ore PPIX might inhibit viral replication.
I am missing a section on the strengths and weaknesses of the study, how reliable is the computational modeling, what might be different reasons for changes in the viral binding properties? What about pH changes due to the different components, indirect effects on cytokine expression within the cell culture, etc.? How will tertiary protein structures and folding characteristics be affected by different compounds? Is the low number of independent experiments genuinely representative? Is in vitro and silico comparable to the human in vivo situation?
R: We appreciate the work and effort of the reviewer who brought essential corrections to the text. We added the suggestions in the Discussion.
Abstract: For me, referring to the high economic burden of COVID19 does not fit the content of the article. I suggest starting with the hematological effects of COVID, which might be explained by your work.
R: We appreciate the reviewer's suggestions. Changes were addressed to the Abstract.
Methods: As far as I understand, Vero cells are isolated from kidney epithelial cells extracted from a monkey. Why have you chosen this kind of cell? Please discuss different cellular approaches in COVID research.
R: Vero cells are a well-established culture for viral replication and are commonly used by many groups that work with Sars-CoV-2 due to their reproducibility and high yield of viral particles. We choose Vero cells because the focus of the work was to produce a high yield of viral proteins for heme/Hb interaction assays.
Discussion: Would it be possible to structure the discussion more by referring to the different results step by step?
R: As suggested, we changed the discussion.
Line 77: You seem to define protoporphyrin IX (PPIX) as heme, but I thought that only the chelation of PPIX with iron forms heme (iron PPIX). For this conversion, the mitochondrial enzyme ferrochelatase is needed; please clarify.
R: The referee is right; PpIX is different from heme. We added this sentence to avoid confusion:
Line 152. “Potent antiviral activity of the heme precursor, PpIX and verteporfin in the nanomolar range has….”
Line 37/38 first sentence "in addition" second sentence "additionally," perhaps it could be rewritten without redundancy.
R: Change was addressed to the manuscript.
“In addition, these compounds and PpIX block the virus's adsorption and replication. Furthermore, we also identified heme-binding....”
4.7. Why have you chosen 10uM heme to be added to 20ug of protein extract?
R: High concentrations of heme (20–50 uM) found in the plasma of patients with severe hemolytic (Roumenina et al., 2018). We choose an intermediate concentration for spectrophotometric studies since it is a sensible technique that allows us to measure changes in the heme spectrum.
Roumenina, L. T., Rayes, J., Lacroix-Desmazes, S., & Dimitrov, J. D. (2016). Heme: Modulator of Plasma Systems in Hemolytic Diseases. Trends in molecular medicine, 22(3), 200–213. https://doi.org/10.1016/j.molmed.2016.01.004
101/102: Why have you chosen the preincubation with 300 μM hemin?
R: Qualitative observation of heme-binding proteins in-gel is not a sensible technique that depends on the concentration of heme-binding protein in a total cell extract, which commonly is low. We tested lower concentrations of hemin but achieved the best result with the saturated concentration of hemin 300 uM.
Figure 3: How often has the experiment been repeated, why are you sure there is no unspecific binding? There is a third line with Hp showing at least one band. Could you specify?
R: The experiments were repeated at least twice, and the new data including a positive and negative control. Indeed, there is a fainted band in the Hp line, possibly a heterodimer of the Hp beta chain (40 kDa) and alpha chain (16 kDa).
Line 288: Please specify whether you used human adult hemoglobin and where it was isolated from (or purchased)
R: Human adult Hb was purchased from Sigma-Aldrich and added a sentence in methods.
“…. solution of 10 µg/mL of human adult hemoglobin (Sigma-Aldrich)….”
Fig. 5: What is the reason for performing only n=2 in the case of Hb-inhibition experiments?
R: We performed different assays and the respective experimental ''n" is now 3. We updated the main text and legends with the recalculated values of inhibition percentages and SD.
Line 78, 82, 342, 344, 345, 350, 457, 474 The citations appear in a different style without brackets
R: Changes was addressed to the manuscript
469: Do you refer to heme-oxygenase 1 (it is written 11)?
R: Change was addressed to the manuscript
Line 564: I guess it should be 5x 105
R: Change was addressed to the manuscript
Figure 5 displays a standard deviation based on two values. Isn’t this an overestimation of results?
R: The standard deviation can be calculated with two replicate values. We agree that three replicates would strengthen the results. We decided to perform additional assays and the respective experimental ''n" is now 3. We updated the main text and legends with the recalculated values of inhibition percentages and SD.
